# A soft selective sweep during rapid evolution of gentle behaviour in an Africanized honeybee

Arian Avalos [1], Hailin Pan[2,3,4], Cai Li[3], Jenny P. Acevedo-Gonzalez[5], Gloria Rendon[1,6], Christopher J. Fields [1,6], Patrick J. Brown[1,7], Tugrul Giray[5], Gene E. Robinson[1,8,9], Matthew E. Hudson [1,6,7] & Guojie Zhang[2,3,4]

Highly aggressive Africanized honeybees (AHB) invaded Puerto Rico (PR) in 1994, displacing gentle European honeybees (EHB) in many locations. Gentle AHB (gAHB), unknown anywhere else in the world, subsequently evolved on the island within a few generations. Here we sequence whole genomes from gAHB and EHB populations, as well as a North American AHB population, a likely source of the founder AHB on PR. We show that gAHB retains high levels of genetic diversity after evolution of gentle behaviour, despite selection on standing variation. We observe multiple genomic loci with significant signatures of selection. Rapid evolution during colonization of novel habitats can generate major changes to characteristics such as morphological or colouration traits, usually controlled by one or more major genetic loci. Here we describe a soft selective sweep, acting at multiple loci across the genome, that occurred during, and may have mediated, the rapid evolution of a behavioural trait.

[1] Carl R. Woese Institute for Genomic Biology, University of Illinois at Urbana-Champaign, Urbana, IL 61801, USA. [2] State Key Laboratory of Genetic Resources and Evolution, Kunming Institute of Zoology, Chinese Academy of Sciences, 650223 Kunming, China. [3] China National Genebank, BGI-Shenzhen, 518083 Shenzhen, Guangdong, China. [4] Centre for Social Evolution, Department of Biology, Universitetsparken 15, University of Copenhagen, DK-2100 Copenhagen, Denmark. [5] Departamento de Biología, Universidad de Puerto Rico, Río Piedras, PR 00931, USA. [6] High-Performance Computing for Biology (HPCBio), Carver Biotechnology Center, University of Illinois at Urbana-Champaign, Urbana, IL 61801, USA. [7] Department of Crop Sciences, University of Illinois at Urbana-Champaign, Urbana, IL 61801, USA. [8] Department of Entomology, University of Illinois at Urbana-Champaign, Urbana, IL 61801, USA. [9] Neuroscience Program, University of Illinois at Urbana-Champaign, Urbana, IL 61801, USA. Arian Avalos and Hailin Pan contributed equally to this work. Correspondence and requests for materials should be addressed to G.E.R. (email: generobi@illinois.edu) or to M.E.H. (email: mhudson@illinois.edu) or to G.Z. (email: zhanggj@genomics.cn)

In 1956, the escape of experimental colonies of an African subspecies of honeybee (*Apis mellifera scutellata*) in Brazil led to broad scale interbreeding with the local European-derived honeybees (EHB)[1]. This resulted in the infamously aggressive, admixed, invasive, New World hybrid population of Africanized honeybees (AHB). AHB achieved its current intercontinental distribution within 30 years, and is now commonly found in the Neotropic and southern ranges of the Nearctic[2, 3]. Aggression of AHB towards humans is well documented[4] and has caused several deaths and widespread public concern in many geographic locations previously dominated by EHB[2, 4]. In 1994, highly aggressive, invasive AHB was first detected in the Caribbean Islands, in Puerto Rico[5].

A remarkable characteristic of the Puerto Rico invasion is that within ca. 12 honeybee generations 1994–2006, the highly aggressive founder AHBs had undergone a drastic reduction in aggression[6], resulting in gentle AHB (gAHB). Current levels of aggression in the Puerto Rico gAHB population resemble those observed in EHB. However, gAHB retains other traits typically associated with AHB, e.g., morphometric dimensions, *Varroa* parasite removal behaviours, and faster queen development[6].

Aggression is generally polygenic[7] and its adaptive value dependent on environmental interactions, making it unlikely that a simple selective sweep can explain rapid evolution of this complex behavioural trait. Rapid phenotypic change on similar timescales to the evolution of gentleness in gAHB has been widely documented across a spectrum of organisms, yet the mechanisms mediating these changes differ from classical models of selection[8–12]. Mounting evidence suggests that admixture[13] or selective sweeps at one or more loci[14] likely mediate such rapid changes in phenotype. However, few examples are known of rapid evolution in behavioural traits, and without genomic analysis it is not clear how such a complex trait might undergo rapid change outside the context of selective breeding. Because they depend on the combined, possibly epistatic interactions of many gene products[7], many of which may have multiple functions,

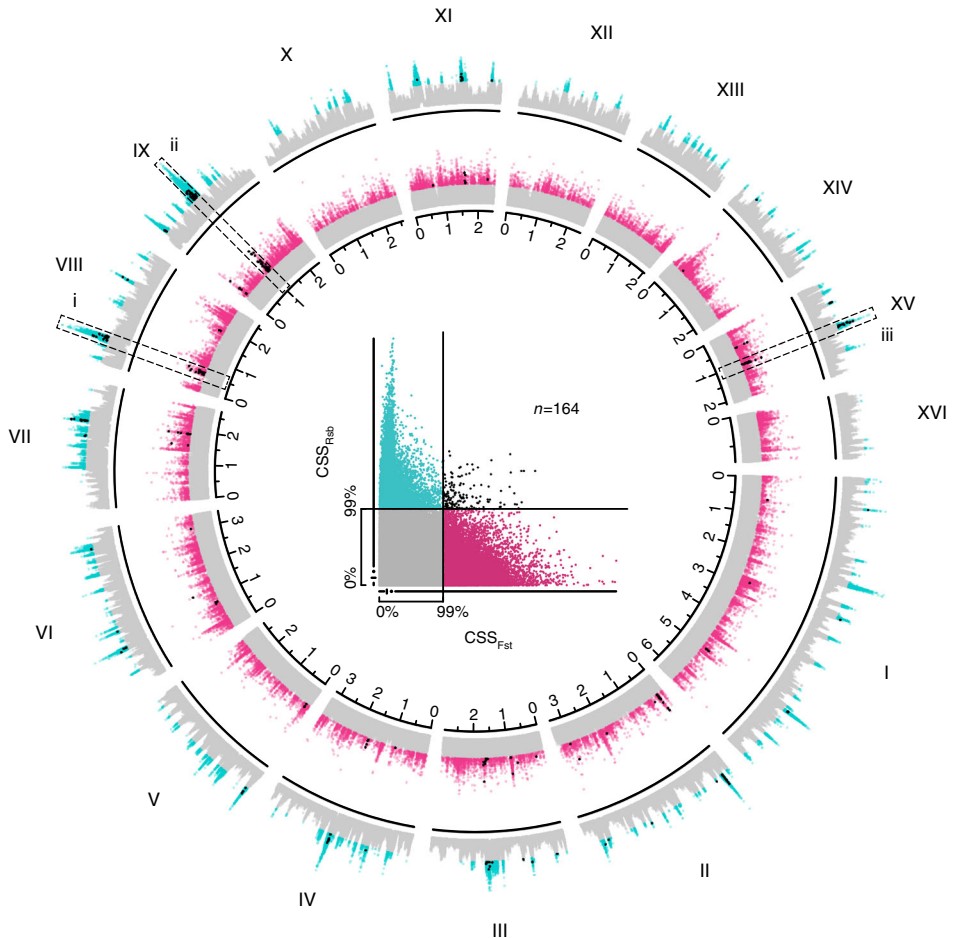

**Fig. 1** Honeybee genome SNPs showing signatures of selection in gAHB. In the central scatter plot, candidate genomic regions associated with the evolution of reduced aggression in gAHB are identified by the distributions of composite selection scores for *Rsb* (cyan + black points) and $F_{ST}$ (violet + black points) at SNPs from genome sequencing of AHB, gAHB and European honeybees (EHB). Axes and cut-off points identify the 0 and 99th percentiles of data distribution; points beyond the 99th percentile were designated as extreme values of interest. The bar along each axis is a reduced box plot with the line extending from the minima to the maxima of the distributions. The gap in the line corresponds to the range from the 25th to the 75th percentile, with the crossbar indicating the median and the point indicating the mean of the distribution. 'gAHB selection alleles' (black points) were defined as the intersection of significant alleles from both the composite selection scores, indicating both an EHB-like allelic profile and significant positive selection unique to gAHB. The radial plot illustrates the distribution of composite selection scores for both metrics across the honeybee genome. Colors identify outliers in the composite score distribution for $F_{ST}$ (violet + black) and *Rsb* (cyan + black) with 'gAHB selection alleles' (black) representing the intersection shown in the center plot. The y-axis extends from the minimum (0) to the maximum composite score value for each metric while the x-axis provides the genetic position in Morgans. Roman numerals identify linkage groups in the honeybee genome. Many significant peaks are apparent across the genome; the dashed boxes and italicized numerals highlight three candidate regions whose component genes are further explored in Fig. 2

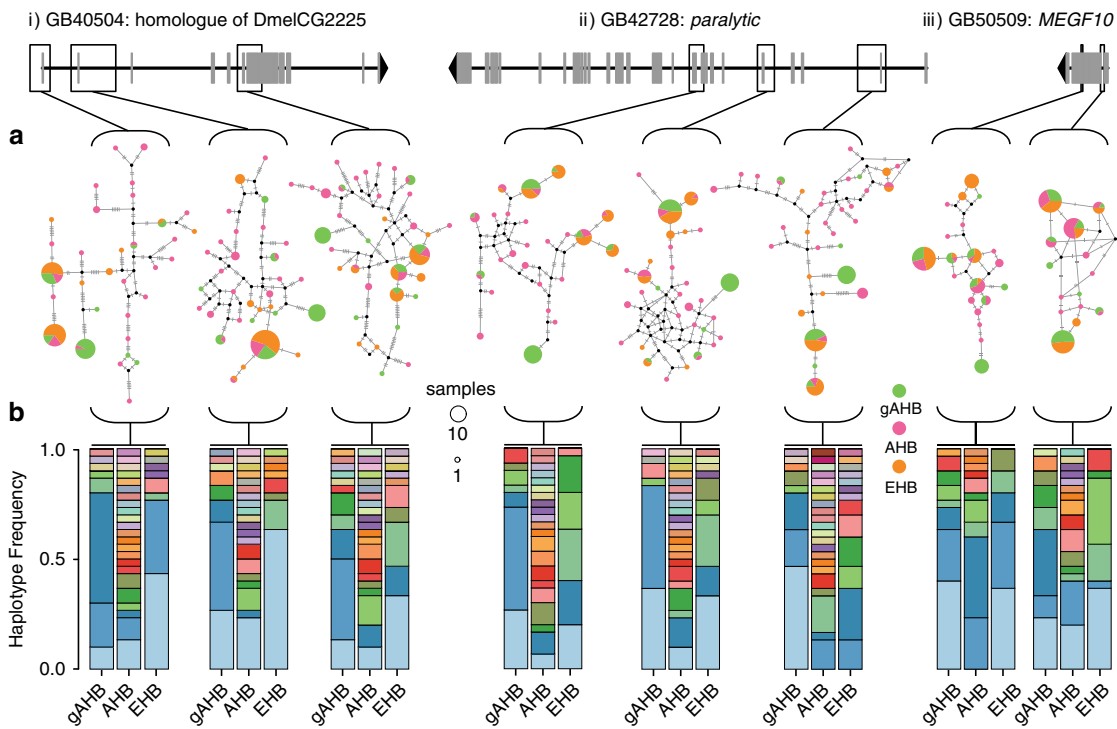

**Fig. 2** Haplotype relationships within three loci under selection in gAHB. We examined haplotype blocks that overlap with exonic regions of three representative genes: GB40504, GB42728, and GB50509. Models are provided for each gene with a black line representing the genomic span, gray segments denoting exons, and an arrowhead identifying 3′ direction. Italicized Roman numerals beside each gene label correspond to regions identified in Fig. 1. Open boxes highlight candidate haplotype blocks within the genes. **a** Median joining network[59] analysis was used to reconstruct the relationships of haplotypes within each haplotype blocks. Circle size is scaled to abundance of each haplotype across the populations, and colors demonstrate the proportion of the given haplotype originating from each population (AHB, EHB, and gAHB). Black dots represent median vertices in the network cross lines represent number of changes between networks. **b** Visualization of the spectrum of haplotypes corresponding to each of the illustrated networks. Each haplotype block is represented by a trio of bar plots for which the y-axis corresponds to the frequency of constituent haplotypes across gAHB, AHB, and EHB populations (x-axis). Constituent haplotypes in each block have unique colors and colors shared across populations indicate the presence of the same haplotype

behavioural phenotypes are likely to require more complex models than those where single loci confer clear selective advantages in isolation, and are ultimately swept to fixation when under sustained selection.

To understand how this remarkable decrease in aggression evolved so rapidly, we sequenced whole genomes of gAHB from Puerto Rico, AHB from Mexico, and EHB from the U.S. ($n = 30$ from each of the three populations) at an average of ×20 coverage. This resulted in 2,808,570 SNP variants, which were characterized within and between populations. We took advantage of the honeybee's haplodiploid sex determination system and sequenced only haploid (male) genomes, to eliminate ambiguity in the haplotype phase. We hypothesized that the genomic regions under selection unique to the evolution of gAHB would be those that differ from AHB, exhibit a more EHB-like allelic profile, and are under positive selection unique to the island population. To simplify our analysis, we investigated whether the decrease in aggression in gAHB was accompanied by altered frequency of the same alleles as those under selection during the evolution of EHB, which also originated from aggressive populations in Africa in the Pleistocene[15, 16] This assumption allowed us to develop a statistically powerful, triangulated analysis.

Our approach identified genomic regions under selection in gAHB that also show EHB-like allelic profiles. These are candidate loci for the EHB-like aggression of gAHB. Haplotype block analysis of these regions showed many haplotypes not found in either of our EHB or AHB populations rapidly became common in gAHB, but that few haplotypes in any regions under selection

were fixed in the gAHB population. We conclude that a soft selective sweep across many loci in the genome accompanied, and may be responsible for, the reduction in aggression experienced by the founding AHB as it evolved towards gAHB.

## Results

**Signatures of selection.** We used extended haplotype homozygosity ($Rsb$)[17] to identify highly localized signatures of greater genetic linkage within the gAHB population compared to the AHB or EHB populations, i.e., regions of selection in the genome (Fig. 1). The method identified 28,086 SNPs as under selection, located in a total of 250 genes. The reciprocal best hit *Drosophila melanogaster* homologs for these 250 honey bee genes (where available) were used against a background set of all *D. melanogaster* reciprocal best hits identified in the honeybee genome (7073 genes) for a Gene Ontology (GO) analysis using the DAVID Gene Ontology bioinformatics database[18, 19]. Several GO categories were significantly overrepresented in this set of 250 genes in regions of positive selection when compared to the universe of 7073 homologous *D. melanogaster* genes (Supplementary Table 1).

This analysis was complemented by a contrast of pairwise comparisons using fixation index ($F_{ST}$)[20], which identified SNPs with extreme $F_{ST}$ values in the gAHB vs. AHB, and EHB vs. AHB comparisons, but lower values in the gAHB vs. EHB comparison (Methods, Supplementary Note 4). Analysis of $F_{ST}$ signal alone showed that overall, few loci approached fixation in any of the

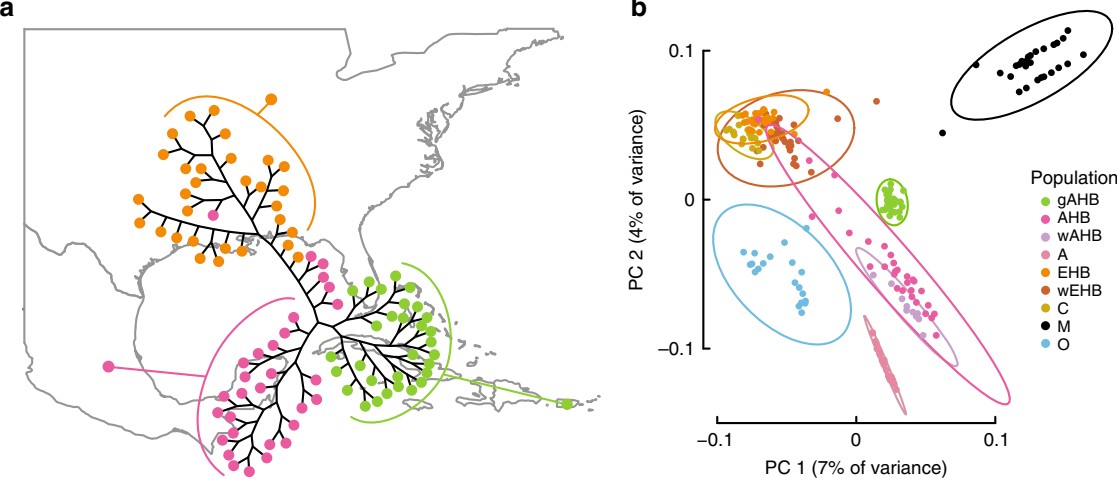

**Fig. 3** Population structure and patterns of admixture across EHB, AHB, and gAHB. **a** A diagrammatic superimposition of an un-rooted Neighbor Joining (NJ) cladogram of the samples with population geography. The NJ tree was constructed from 2,808,570 SNPs derived from whole-genome sequencing. Taxa represent the 90 samples assessed in this study, with fill color corresponding to population membership. **b** Principal component analysis (PCA) showing population structure of 230 samples with 1,049,512 SNPs common between the 90 samples in **a** and 140 previously-described samples of wild and domesticated honeybees from Europe, Africa, and North and South America[29]. Ellipses in the PCA enclose the 95th percentile of actual population membership with ellipse color corresponding to population as above. The gAHB, AHB, and EHB group labels are the populations sampled in this study. Remaining labels are aggregates of samples from the previous study[29] grouped as follows: wAHB is a Brazilian AHB population; wEHB are domestic EHB population samples from both Minnesota, USA, and Sweden; groups A, O, M, and C are genetic groupings spanning multiple, distinct subspecies of *A. mellifera* as identified in prior reference studies[15, 29]

individual comparisons. Lower $F_{ST}$ values were detected in the gAHB vs. AHB comparison than other comparisons, providing further evidence for an AHB origin of gAHB (Supplementary Fig. 1). Application of the $F_{ST}$ contrast further narrowed the SNP candidates by highlighting those with an EHB-like allelic profile that are under selection in gAHB. The loci showing the strongest signatures of selection by *Rsb* also contained many SNPs with EHB-like allelic profiles and significant $F_{ST}$ scores (Fig. 1). We interpreted this as the gAHB population evolving a more EHB-like profile at selected loci, thus representing loci that may be associated with the evolution of gentle behaviour. By calculating composite selection scores (CSS)[21] and isolating SNPs with extreme scores in both the $F_{ST}$ and *Rsb* metrics, we arrived at 164 SNPs with signatures of selection in the gAHB population (Fig. 1).

**Haplotype analysis and gene annotation**. Many of the candidate SNPs with extreme values in both metrics co-localized to the same genes. We therefore conducted haplotype block identification to capture broader signals in the region around selected SNPs (Methods, Supplementary Figs. 2 and 3). Using GERBIL[22] we identified 216,655 haplotype blocks across the honey bee genome with 128 of these containing at least one of the 164 SNPs of interest. As expected, because of the honeybee's highly recombinant genome[23], haplotype block sizes were small, with the majority of blocks spanning < 5000 base pairs (Supplementary Fig. 2) and containing 30 SNPs on average (Supplementary Fig. 3). Of the 128 haplotype blocks we identified, 56 were in intergenic regions, possibly indicating regulatory sequences subject to selection pressure. Exons or introns of 35 protein-coding genes contain the remaining 72 selected blocks. Twenty of these genes have known homologs in *D. melanogaster* and several have been implicated in the regulation of honeybee aggression[24]. They are involved in a variety of processes including transcriptional regulation, protein modification and transport, cell adhesion, and chromatin binding (Supplementary Data 1).

We used haplotype network analysis to investigate whether selection at the 128 haplotype blocks favored EHB haplotypes in gAHB, or whether distinct processes governed the loci under selection in gAHB (Fig. 2a). Results showed that certain haplotypes present in both the AHB and EHB populations appear to have become more frequent in gAHB (Fig. 2b). In addition, some haplotypes present at high frequencies in gAHB are unique to this population as sampled (Fig. 2). Our analysis revealed: (1) extensive haplotype variation in AHB at these loci, greatly exceeding that found in the other two populations, (2) gAHB haplotypes that coincide strongly with EHB haplotypes in some but not all cases, and (3) evidence for either the evolution of gAHB specific haplotypes or greatly increased frequency in gAHB of rare haplotypes not detected in our sampling of EHB and AHB (Fig. 2). These results further support a soft selective sweep event in gAHB distinct from the evolution of EHB, in which multiple haplotypes at multiple loci jointly reached similar frequencies.

**Analysis of population diversity and selection using the sex determination locus**. We also employed an analysis of variation at the *csd* locus as a proxy for population diversity[25]. Allelic combinations of *csd* are known to be the primary mechanism for sex determination in honeybees[26, 27]. Homozygous individuals at the *csd* locus result in diploid males, a costly by-product of this system, as they produce sterile females (queens) and are thus killed by workers[25, 28]. Recently it was shown that balancing selection at *csd* has been a key evolutionary driver in the invasion of *Apis cerana* in Australia[25]. This evidence suggests that the distribution of *csd* alleles in a population can be used to assess diversity in *Apis* populations in the context of biological invasions.

The allelic profile of gAHB was used to determine whether balancing selection favouring rare *csd* alleles was a driving component in the evolution of gAHB, and whether it could explain the high genetic diversity still present in the population. We used *P(d)* (Methods) to calculate a likelihood metric for the

diploid males in the subsequent generation to the haploid gAHB drones that we sequenced. Of the three populations, gAHB had the highest likelihood of producing diploid drones. Almost twice the likelihood of a diploid male was predicted in gAHB than in the AHB population sampled, and more than three times the diploid male likelihood of the EHB population sampled. gAHB has a lower total number of *csd* alleles, supporting a bottleneck effect that has been maintained for ~10 years. In addition, the *csd* alleles in gAHB show very unequal frequencies. If balancing selection had been a driving factor in the evolution of gAHB (as in the Australian *A. cerana*[25] population), the expectation would be that multiple alleles at similar frequencies would be prevalent in the population. One allele is by far the most frequent in gAHB (Supplementary Fig. 4), making this unlikely.

### Relationships between gAHB and other honey bee populations.

Phylogenetic analysis revealed that gAHB is a monophyletic group likely derived from a founder population on the AHB-EHB spectrum (Fig. 3a). Consistent with this, clustering analysis revealed distinct AHB-like ($n = 56$) and EHB-like ($n = 34$) clusters (Supplementary Fig. 5), surprisingly while all gAHB samples clustered within AHB, some AHB ($n = 4$) samples were consistently assigned to the EHB cluster (Fig. 3a, Supplementary Fig. 5b). To further ground the genomic profile of gAHB within known honey bee populations we utilized a previously published whole genome data set[16]. The intersection of the two sets contained 1,049,512 variant calls representing 60% of the Wallberg et al.[16] SNP set and 37% of the SNPs examined here. Principal component analysis of the intersection further confirmed that gAHB is most closely related to the admixed Mexican AHB population, suggesting gAHB origin from AHB on the North American continent (Fig. 3b). The previously described[16] AHB population from Brazil (wAHB; Fig. 3b) shows substantially less admixture with the EHB + C Group than the Mexican AHB population. Recent studies also indicate there may be greater contributions from the M Group in the Brazilian population[29]. We did not observe the pattern expected if admixture of gAHB with EHB had occurred on the island after the invasion event and bottleneck. Instead, the gAHB population forms a well-defined cluster that supports origin from an aggressive AHB founder population closely related to Mexican AHB.

### Discussion

Here we show that gAHB evolved via selective pressure on standing variation within an admixed North American AHB founder population that arrived on Puerto Rico. Genomic regions showing signatures of selection in gAHB (Fig. 1) contain multiple haplotypes at approximately equivalent frequencies (Fig. 2b). This is consistent with the $F_{ST}$ results, which showed that while signatures of selection are present, few alleles are truly fixed in any of the three populations (Fig. 1, Supplementary Fig. 6). Multiple haplotypes with similar frequencies and low degrees of fixation are characteristics of soft selective sweeps[9, 30, 31]. We therefore conclude that a soft selective sweep occurred in gAHB, and that this accompanied and perhaps mediated a rapid change in behaviour across the population.

Our results contrast with the evolution of decreased aggression by artificial selection during vertebrate domestication[32]. In domestication, extreme selection pressure and control of pedigree generally leads to hard selective sweeps, and decreased aggression is often associated with retention of juvenile characteristics together with color and morphology changes in a domestication syndrome[33]. In gAHB, aggressive behaviour in colony defense showed rapid evolutionary change but other characteristics such as morphology, queen pupation rate, and aggression against

parasites remained unchanged[6]. Although predecessors of the EHB population also evolved from an African precursor most similar to the current A genetic group[15], the distinct population genetics of gAHB relative to all domesticated bees (Fig. 3b) shows that gAHB and EHB arose by separate events. Gentle honeybees have thus arisen multiple times, from different founder populations, by related but distinct population genetic mechanisms, and with at least some similarities in genetic architecture.

The selection pressure that led to gentleness in gAHB maintained both genetic diversity and other AHB characteristics. Our *csd* analysis suggests that this occurred despite a genetic bottleneck. We did not observe the evidence of balancing selection in *csd* expected if sex determination was a driver of gAHB evolution. Also, the high likelihood of diploid male production in gAHB suggests that the cost of the genetic bottleneck is outweighed by the selective advantages of gAHB.

The forces driving the evolution of gAHB are challenging to determine conclusively. We propose that a combination of three factors, negative human–honeybee interactions, geographic isolation, and low levels of predation on honeybee colonies, may have driven selection against honeybee aggression in Puerto Rico, which is presently the only place in the New World where gentle Africanized honeybees have evolved. First, the human population density in Puerto Rico is the highest within the range of any AHB population in the New World[34]. Destruction of highly aggressive colonies in the expanding AHB population by humans early in the invasion period was likely more frequent than in other New World ecosystems, acting as one key selective force. Second, geographical isolation of Puerto Rico would have increased the selective potency of these negative human–honeybee interactions. The island is remote enough to present serious barriers to natural insect dispersal, particularly to a swarm-founding species such as the honeybee, limiting escape. Third, Puerto Rico does not have major vertebrate[35] or invertebrate[36] colony-level predators common elsewhere in the range of AHB, likely relaxing selection favouring aggression in feral colonies. Thus, selection likely eliminated the most aggressive of the invasive AHB colonies. With reduced competition for floral resources, gentler colonies would have increased reproductive success. In addition, reduced aggression may also have served as an exaptation, enabling exploitation of nesting sites and resources in urban landscapes inaccessible to the more aggressive AHB colonies. These factors would effectively increase the frequency of certain AHB haplotypes at the expense of most others (Fig. 2). Some of these haplotypes, in combination, presumably confer gentle behaviour.

According to this scenario, the founder AHB population would have experienced an initial strong selective pressure towards reduced aggression, which would stabilize as gentle colonies became predominant and instances of negative interactions with humans decreased. High recombination rates in honeybees make selection for multiple loci extremely efficient and may facilitate the speed of evolution in honeybees.

Although we favor initial pressure from unsupervised human-driven selection as a primary driver in the evolution of gAHB, alternatives exist. One possibility is that the founding AHB colony happened to be biased towards haplotypes conferring gentle behaviour, which are now over-represented in gAHB as a result of drift in the founder population followed by a genetic bottleneck. It could be the case that, by chance, the founding AHB colony contained a relatively high proportion of alleles conducive to gentleness. All available reports[6] indicate that the founding AHB population in PR was aggressive, and that the reduction in aggression characteristic of gAHB arose after the invasion. Aggression in all other AHB invasion events has been shown to be adaptive and dominant[1, 4], hence the expectation in PR was a retention of aggressiveness even if associated haplotypes were

initially present in low frequencies. This has not been the case, suggesting a countering factor. We conclude that selection for gentle behaviour, or associated traits, is this factor. Further, the signatures of selection identified here via allele frequency are corroborated by Rsb (i.e., linkage disequilibrium) measures. As linkage disequilibrium is rapidly lost in honey bee populations due to their extreme recombination rates, retention of linkage is a strong indicator of recent selective pressure. Finally, we observe haplotypes at loci under selection in gAHB that are rare or absent in the presumed AHB source population.

Another factor that could be related to the evolution of gAHB is the constraint presented by resource cycles in tropical oceanic islands[6, 37]. According to this scenario, selection towards a more EHB-like resource acquisition strategy may have led to the current gAHB. This could only occur if either the same underlying traits and loci are involved in resource acquisition and aggression, if these loci are linked genetically, or by epistatic interactions. This speculative explanation would also be consistent with the observed soft selective sweep, retaining AHB haplotypes that likely confer aggressive traits in the population.

Our findings have implications for understanding both rapid evolution and the genetics of biological invasions in colonial organisms. We show that haplotypes not found in our EHB or AHB samples rapidly became common in gAHB. These haplotypes were either present but rare in the founder population, or emerged after the invasion event. Rapid emergence of new haplotypes in honeybees is likely, due to their following genetic characteristics: high within-colony genetic diversity[38], extremely high recombination rates[23], and extensive outcrossing and thus gene flow between colonies[39]. Some newly emerged haplotypes may be the key to the separation of aggression at the colony level from other traits that was observed for the first time in gAHB. The future events in this selective process are uncertain. One possibility, consistent with classical evolutionary theory, is that the observed soft selective sweep would harden as the evolutionary process continued, fixing the less aggressive behaviour in the population. However, colony-level selection and haplodiploid outbreeding likely affect the population dynamics even of highly favorable alleles in honeybees in ways that may promote soft over hard selection over longer timescales. We suggest that these factors provide a selective advantage to the species as a whole by enabling multiple, sequential adaptive radiations. Such an advantage in invasive situations could explain the overall success of the haplodiploid system despite its very high biological cost[25].

It is also noteworthy that gentle honeybees that are genetically distinct from the extant EHB populations can evolve rapidly from a small founder population of AHB. This is an important result as EHB in particular are principal pollinators of many domesticated plant species, but they are under threat from multiple sources, endangering production of many important agricultural and horticultural crops. Genetically diverse gentle honeybees could help secure agricultural production by providing pollinators more resistant to threats such as parasites and diseases. The evolution of gAHB also provides a paradigm for the adaptation of social organisms to new environments. The ability of eusocial insects to generate and maintain diversity within a colony at loci mechanistically involved with complex traits such as behaviour may be a key to their evolutionary success.

## Methods

**Sample collection and sequencing**. A single male (drone) honey bee was collected at the pupal or recently eclosed stage from each of 90 unrelated colonies across three major geographic locations. Thirty gAHB samples were collected from apiaries across Puerto Rico and adjacent islands of Vieques and Culebra, which are known to be part of the breeding population[40]. In total 30 EHB samples were collected from research apiaries maintained by the University of Illinois Bee Research Facility in Champaign County, Illinois. The remaining 30 AHB samples

were collected from the research apiaries maintained by the Centro Nacional de Investigación Disciplinaria en Fisiología y Mejoramiento Animal, a member research division of the Instituto Nacional de Investigaciones Forestales, Agrícolas y Pecuarias (INIFAP) in Querétaro, Mexico. Individuals no younger than the white-eyed stage of pupal development were selected to assure unequivocal identification as a drone. Later in the season (October–November) when drone production slowed, we specifically targeted drones that had recently eclosed. All bee specimens were preserved in 95% ethanol upon collection and transported to research facilities at the University of Puerto Rico, Rio Piedras for DNA extraction and processing.

Samples were processed by separating the thorax from the rest of the body and split into two equal parts. One half of the thorax was stored as archival material, while the other half was washed with molecular grade water (item W4502, Sigma-Aldrich St. Louis, MO) to remove excess ethanol and placed in a micro-centrifuge tube in a container with ice. Sample DNA extraction was done using the QIAamp DNA Mini Kit (QIAGEN Germantown, MD) with minor modifications to the manufacturer's protocol. Specifically, extraction method diverged from protocol at the end of the extraction, where two elution washes of 40 µl of buffer AE (from kit) were conducted rather than the one recommended wash. Resulting DNA quality and quantity was measured using three methods: agarose gel electrophoresis (1%), Nanodrop (NanoDrop ND-1000), and Qubit Fluorometer (per the manufacturer's instructions).

Following DNA extraction, samples were shipped to BGI facilities in Tai Po, Hong Kong, where 500 bp insert-size libraries were prepared for each sample per manufacturer's directions and sequenced using the Illumina HiSeq 2000 platform. Extracted DNA was fragmented via Covaris sonicator, and quality assessment followed using Gel-Electrophotometry. Fragmented DNA was combined with End Repair Mix and incubated at 20° C for 30 min. We followed up this step with purification using the QIAquick PCR Purification Kit (QIAGEN), then added A-Tailing Mix and incubated at 37° C for 30 min. Adapters were added to this adenylated DNA mix using Adapter and Ligation Mix and an incubation reaction at 20° C for 15 min. Size selection was done on a 2% agarose gel to recover the target insert size (500 bp), followed by a gel purification step via QIAquick Gel Extraction kit (QIAGEN). Several rounds of PCR amplification with PCR Primer Cocktail and PCR Master Mix were performed to enrich the fragments. The PCR products were further selected by running another gel purification step to recover the target fragments. The final library was quantitated in two ways: (1) determining the average molecule length using the Agilent 2100 bioanalyzer instrument (Agilent DNA 1000 Reagents), and (2) quantifying the library by real-time quantitative PCR (qPCR) (TaqMan probes). Upon quantitation, high-quality libraries were paired-end sequenced using the Illumina HiSeq 2000 platform with 100 bp read length.

**Variant calling**. GATK best practices[41] were used to analyze the 90 haploid honeybee drone samples (Supplementary Fig. 6). All analysis steps were applied except for INDEL recalibration (which requires a set of known variants).

Prior to input, raw read files were quality checked and trimmed with Trimmomatic version 0.33[42]. As part of the process quality scores were first converted to Sanger format (TOPHRED33 option) then processed by clipping the remaining adapter sequences (TruSeq2-PE.fa:2:30:10:6:true options). Reads were trimmed at the leading and trailing ends if below the threshold quality score (28), and in addition a sliding window trimming approach was applied (sliding window size 4 bp, quality score threshold 15). Lastly, reads smaller than 30 bp post-trimming were excluded.

Trimmed reads were aligned to the honeybee reference genome (BeeBase, scaffold assembly of Amel 4.5) with BWA MEM[43] (version 0.7.10) using -M parameter. Aligned/properly mapped reads were de-duplicated with SAMBLASTER[44] (version 0.1.22). De-duplicated samples were then realigned with GATK[45] (version 3.4-0) RealignerTargetCreator followed by IndelRealigner (see Supplementary Note 1, Step 1; Supplementary Data 2). Raw variants were calculated for each realigned sample using GATK[45] (version 3.4-0) HaplotypeCaller (see Supplementary Note 1, Step 2). The variant calling process resulted in individual **.gvcf files which were subjected to joint variant calling using GATK[45] (version 3.4-0) GenotypeGVCFs (see Supplementary Note 1, Step 3).

**Variant filtering**. Initial SNP calls that numbered 8,706,689 were further filtered by quality and representation. The first set of filters addressed (1) potential artifacts due to current reference assembly architecture and (2) reliable representation of SNPs across the data set. Unplaced Scaffolds were first excluded from our analysis, as they are regions prone to false SNP calls due to similarity or overlap with other scaffolds[46]. Removal was achieved by generating an index file for all "GroupUn" scaffolds using base packages in R[47] and then applied via VCFtools version 0.1.14[48] (see Supplementary Note 2, Step 1).

Regions of nuclear insertions of the mitochondrial genome (NUMTs) were also excluded from our data set as there are several highly similar NUMTs in the honeybee genome[49, 50]. These regions were identified in the current assembly using NUCmer version 3.0[51, 52], and regions with >80% identity were re-formatted as a **.bed file, then excluded (see Supplementary Note 2, Step 2).

Quality filters were applied to the resulting **.vcf using GATK[45] –VariantFiltration and –SelectVariants (see Supplementary Note 2, Step 3). A

minimum allele frequency filter (MAF) was then also applied, removing all SNPs with < 5% frequency across all 90 samples. Analysis was further constrained to bi-allelic SNPs; 194,689 sites with multiple alternative allele calls (>1) were detected in the data set and excluded. Individual, manual assessment of these 194,689 sites showed that they may have resulted from possible alignment errors across gaps or INDEL events and thus removed from analysis. Lastly, we validated the degree of overlap between our final 2,808,570 SNP data set and a previously published, whole-genome sequencing data generated by Wallberg et al.[16].

**Examination of genetic groupings.** Genomic profiling of populations was conducted by taking advantage of previously published data sets. In particular, a whole-genome sequencing data set of diploid female worker honey bees from Old-World populations conducted by Wallberg et al.[16] was used to anchor the present study's three closely related populations within a broader context of honey bee genomic variation. The Wallberg et al. data set was first filtered in the same manner as above, except that the per-sample representation filter (AN < 72.0) was not applied, and different read depth (DP < 269.2685 || DP > 1484.405) and quality filters (QUAL < 15.776 || QUAL > 8072.961) were implemented to represent data-set specific quality analysis results. The compiled population data set, i.e., samples from this study and those from the reference study[16], was used in a principal components analysis (PCA) (Fig. 3).

Genetic groups specific to our three populations were examined using a sequential k-means clustering approach[53]. The analysis was conducted using the adegenet package in R[53–55]. The approach first applies dimensional reduction via PCA using the glPca function to the SNP data set then conducts successive K-means clustering using the find.cluster function in the package. This successive process was performed with increasing number of clusters and derived a Bayesian information criterion as a goodness of fit measure for each cluster, this was used to select the optimal k (see Supplementary Note 3).

**Assessment of population diversity through the csd locus.** An analysis of the complementary sex-determiner gene (csd)[26] was applied to assess (1) the effective male population size across all three study populations (EHB, AHB, gAHB), and (2) the likelihood of a founder effect in gAHB. The method used the GERBIL[22] imputed SNP data set (described below) to extract all non-synonymous SNP variants within coding DNA sequence regions of the csd gene. Non-synonymy was determined via annotation with SnpEff version 4.2[56] using the *Apis mellifera* official gene set obtained from BeeBase. The combination of these non-synonymous coding SNPs thus represents allelic variants of the csd gene. A simple (ε = 0) median joining network was computed using the algorithm in PopArt[57, 58] to assess the distribution of alleles across gAHB, AHB, and EHB (detailed below). In addition, for each population, the likelihood of diploid males $P(d)$ was derived using the formula:

$$P(d) = \frac{\sum_{i=1}^{k} \binom{h_i}{2}}{\sum_{i=1}^{K} \binom{h_i}{2}} \quad (1)$$

where $k$ is the number of csd alleles present in each one of the three populations (gAHB, AHB, EHB), $\binom{h_i}{2}$ is the possible number of diploid males containing the allele $h_i$ in the population, and $K$ is the total number of csd alleles present across the three populations.

**Fixation index (Fst).** For $F_{ST}$[20, 59] derivation, a modified function (WC_Haplotype_Fst_Function.R) was applied that calculated haploid $F_{ST}$, as described by Weir[20]. Briefly, the function incorporates as input a list of population membership, and a matrix where SNPs correspond to row entries and samples correspond to columns (see Supplementary Note 4, Step 1). Matrix values must be binary with a 0 representing individuals with the same allele as the reference genome, while a 1 represents the alternate allele as identified in the variant calling step. Population membership should match the order of samples in the matrix. For each pairwise calculation of $F_{ST}$, the matrix was partitioned so that it housed only the two populations of interest (gAHB v. EHB or gAHB v. AHB or AHB v. EHB), then the function WC_Haplotype_Fst_Function.R was applied (see Supplementary Note 4, Step 2). Validation of $F_{ST}$ was achieved by permutations of population membership as recommended by Weir[20]. A matrix housing randomized population memberships was generated, then each row entry was relayed as input to WC_Haplotype_Fst_Function.R in an iterative fashion (see Supplementary Note 4, Step 3).

The number of times a permutation-derived $F_{ST}$ value equaled or surpassed the observed $F_{ST}$ value was divided by the total number of iterations. This ratio equates to a $P$-value specific to the permutation-derived distribution of $F_{ST}$ for each SNP in each pairwise comparison. As the permutation-derived distribution assumes, through randomization of sample membership, that there are no differences between the populations in each pairwise comparison, this $p$-value is a measure of how likely it is that our observed $F_{ST}$ fits that null hypothesis (see Supplementary Note 4, Step 4).

**Extended haplotype homozygosity (Rsb).** Rsb served as an independent assessment of selection[17]. Rsb is a measure of the sequence entropy surrounding individual SNPs. The assumption is that that any deviation from random patterns of segregation in a target population (e.g., selection) reduces the level of entropy and thus increases the degree of association between variants. Implementation of Rsb was used to identify those SNPs that had low surrounding entropy in the gAHB population compared to the AHB or the EHB populations.

Calculation of Rsb used an imputed form of the SNP 0,1 data matrix created during haplotype block derivation using GERBIL (described below)[22]. This prior step allowed the rescue of missing SNP calls which would have otherwise been discarded during the calculation of the Rsb statistic. The imputed SNP matrix was re-formatted to a haplotype data format required by the rehh package in R[47]. This step resulted a mapping file, and 909 haplotype files where each of the 303 scaffolds with detected variation was represented by three files. Each one of the files contained samples from one of the populations (gAHB, AHB, EHB). Data in the haplotype files followed the original matrix orientation with SNPs as rows and samples as columns, but alleles were coded using the nucleotide base pair rather than binary identification (see Supplementary Note 4, Step 5). Once all the files were generated, a looping function was used to iteratively apply the scan_hh function in the rehh packages. The scan_hh function scanned the haplotype files using the mapping file as reference, calculating population- and scaffold-specific statistical precursors to Rsb (see Supplementary Note 4, Step 6). Concatenated output files contained the position information of each SNP (CHR, POS columns), the reference (REF) allele frequency (freq_A), the iHH metric for both the reference and alternate (ALT) alleles (iHH_A, and iHH_D respectively), and two calculations of iES (iES_Tang_et_al_2007[17], iES_Sabeti_et_al_2007[60]).

The resulting files were used as input for the ies2rsb function in the rehh package. The ies2rsb function directly calculated Rsb using the values in the iES_Tang_et_al_2007 entry. The iES_Tang_et_al_2007 values correspond to the integrated area under the haplotype homozygosity curve as the haplotype is extended outwards from each specific SNP in both directions[17]. The median deviation of the logarithmic ratio of iES between two populations for each SNP $i$ (ln $(Rsb_i)$')[17] yields ln$(Rsb_i)$, which is a normally distributed measure of fold change deviation between iES. A positive fold change indicates slower haplotype decay in the dividend population, which corresponds with greater linkage disequilibrium and is a signature of positive selection. Rsb is the exponential of ln(Rsb) bound between 0 and infinity, with values greater than 1 indicating positive selection. The rehh package in R outputs the fold change: ln(Rsb), which was exponentiated to yield Rsb (see Supplementary Note 4, Step 7).

**Composite selection score (CSS).** To compare the $F_{ST}$ and Rsb signatures of selection, a Composite Selection Score (CSS) was implemented as described in ref. [21]. Briefly, a fractional rank is first derived for each individual calculation (e.g., a single $F_{ST}$ pairwise comparison), then converted to a Z-statistic. The resulting Z-score is averaged for each SNP and compared against a normal distribution of Z-scores to derive a $p$-value. The CSS $p$-value represents the degree of deviation from the distribution of Z-scores, which can then be used to detect significant deviations in signal strength or whose inverted form (-Log10 of the $p$-value) can be applied as a granular scale of deviation from normal to detect outliers.

A CSS was derived for each individual metric ($F_{ST}$, Rsb) then contrasted between the two metrics to identify overlapping outliers which are defined here as individuals with CSS scores > 99% of all SNP CSS scores in each metric (Fig. 1). This approach was readily applicable to Rsb as it contrasted gAHB samples to each AHB and EHB samples. It also was applicable to SNPs under positive selection unique to gAHB that would have values greater than 1 (see above description).

In contrast, the CSS calculation for $F_{ST}$ was more nuanced as SNPs of interest for the strategy of pursuing EHB-like allelic profiles in gAHB were those showing signatures of selection in the gAHB v. AHB, and EHB v. AHB pairwise comparisons, while also non-distinct in the gAHB v. EHB pairwise comparison. To address this, the CSS score for $F_{ST}$ was constructed using the direct values of the gAHB v. AHB and EHB v. AHB comparisons together with the mathematical negation of the gAHB v. EHB comparison. In this way, a high CSS $_{FST}$ score represents SNPs that are closer to fixation in the gentle populations relative to aggressive populations while simultaneously showing more similar allele profiles between the gentle populations.

**Haplotype blocks.** It is widely known that honey bees possess extreme rates of recombination[23]. This has historically limited our ability to detect signals of linkage disequilibrium (LD) across the honeybee genome using marker data. Whole genome sequencing provides a greater degree of resolution, and makes it possible to directly assess LD signal across the honey bee genome[61]. The software GERBIL (version 1.1), a tool in the GEVALT software suite[22], was used to capitalize on the greater resolution of genomic markers and aggregate SNPs into discrete haplotype blocks. As GERBIL is only able to use diploid input, the SNP data set was first converted to a homozygous diploid data set, then re-formatted to the format described in the GERBIL software manual. Briefly, individual tab-delimited files were produced for each of the scaffolds, with columns representing SNPs and every two consecutive rows corresponding to one sample. SNPs were represented in 0, 1 format or a "?" if missing. Re-formatted files were analyzed in an iterative fashion

using default software parameters (sample code: gerbil.exe <input filename> <output filename>).

**Haplotype networks**. Distribution of haplotypes within and across the populations was assessed through median-joining networks[58] of the 72 haplotype blocks that overlapped with honeybee genes showing significant signatures of selection as described earlier. As haplotype blocks constitute regions where recombination was low in one or more individuals, component SNPs represent a grouping whose allelic pattern can provide insight into the degree of genetic variation. Networks were constructed using the median-joining network algorithm in Pop-Art version 1.7[57], with the simplest network drawn up for each haplotype ($\varepsilon = 0$)[58].

**Code availability**. A full copy of the custom script R function used for $F_{ST}$ calculation (WC_Haplotype_Fst_Function.R) along with detailed annotations is made available at our project repository (https://github.com/HPCBio/Honeybee-VariantCalling).

**Data availability**. All genomic sequencing data generated by this study is available via the NCBI Short Read Archive repository under the bioproject ID PRJNA381313. The NEXUS format files for of the haplotype blocks used for the haplotype network analysis is available at: https://github.com/HPCBio/Honeybee-VariantCalling. Haplotype networks can be readily re-constructed through upload (File > Open > **.nex) and execution of the median-joining network algorithm (Network > Median Joining Network) within the PopArt (version 1.7)[57, 58] software suite for any one of the files provided. An additional data set implemented in the population structure analysis is referenced in Wallberg et al.[16] (doi:10.1038/ng.3077), and was obtained from the authors with permission. All other data are available from the corresponding authors upon request.

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

## Acknowledgements

We are grateful to G. Diaz, C.D. Nye, C.C. Rittschof, and J. Uribe-Rubio for sample collections; members of the University of Illinois Carver Biotechnology Center High Performance Biological Computing group for assistance with data analysis; and A.M. Bell, A. Zayed, C.W. Whitfield, J. Catchen, K.A. Hudson, and members of the Robinson and Hudson laboratories for comments and suggestions that improved the manuscript. This work was supported by Lundbeck Fellowship to G.Z. (R190-2014-2827), the Chinese Academy of Sciences XDPB0202 (H.P., G.Z.). This work was supported by USDA Farm Bill Grant 16-8272-2014-CA (T.G.), Puerto Rico Science, Technology, and Research Trust grant 2016-00161 (T.G.), NSF-OISE grant 2015-1545803 (T.G., J.P.A.-G.), and an NSF Grant IOS-1256705 to G.E.R. This material is based upon work supported by the National Science Foundation under Grant Number NSF 15-501 awarded to A.A. Any opinions, findings, and conclusions or recommendations expressed in this material are those of the authors and do not necessarily reflect the views of the National Science Foundation.

## Author contributions

G.E.R., M.E.H, G.Z., T.G., H.P. and A.A. contributed to study design, data analysis, and writing the manuscript. J.P.A.-G. conducted DNA extractions and managed sample logistics as they arrived from collection sites and later when sent to BGI for sequencing. H.P., C.L. and G.Z. mediated sample sequencing. G.R., C.J.F. and A.A. developed, tested, and implemented the detailed variant calling pipeline. H.P. and A.A. conducted independent parallel initial analyses of the data which were combined into the approach described in the manuscript. P.J.B. assisted with haplotype analysis, conceptualization, and statistical assessment.

## Additional information

**Competing interests:** The authors declare no competing financial interests.

