## [Peer Review File · Nature Communications]

Reviewers' comments:

Reviewer #1 (Remarks to the Author):

General Comments:

This is an excellent study that uses an extremely large data set (90 independent genomes from three different geographic locations). The combination of F_{ST} and haplotype homozygosity (as a measure for selection) helped identify many candidate regions and genes. I recommend the publication of this manuscript but would ask the authors to include/discuss alternative explanations (see below).

The presentation and documentation of their methods is very good.

Specific comments:

1. I am not sure that the author could rule out an alternative scenario to their proposed hard/soft sweep, namely, colonization by a founding population that was exclusively gAHB or had a high frequency of gAHB genotypes already.

I assume the author mean selection (see below) by humans in an urban environment. Selection likely involved the elimination of the most aggressive of the invasive AHB colonies.

147 With reduced competition for floral resources, gentler colonies would have increased reproductive

148 success.

This didn't happen in any other population where a lot of africanized feral bees are still out-competing the highly managed EHB populations. Is there anything known about the composition of the feral africanized bees. If those are also gentle an alternative explanation would be that the initial colonizers were already gentle or gentle AHB were colonizing the island at a high frequency and this trait was fixed by genetic drift.

In particular, looking back at the published records on the rapid decrease in aggressiveness seems not that conclusive, i.e. the evidence is based on public media reports rather than hard scientific evidence.

Marchand et al. 2012:

In preliminary observations, the Africanized bees of Puerto Rico seemed to be less defensive than expected while retaining their typical size and mite resistance. Moreover, a dramatic decrease in reported attacks since the arrival of Africanized bees in Puerto Rico (Rivera-Marchand 2006) supports the hypothesis of decreased defensiveness. For instance, there had been four deaths in the first 4 years of the arrival of Africanized bees and none in the last 10 years. This pattern is significant in a simple runs test. The number of incidents where bees attacked people, as investigated in newspaper archives, shows a 100-fold decrease in numbers in the same time period. These data may indicate reduced defensiveness in Africanized bees from Puerto Rico, although public awareness could account for part of the decrease in incidents (Rivera-Marchand 2006).

Finally, additional evidence for a unique composition of the island population is given by the authors

In addition, some

102 haplotypes present at high frequencies in gAHB are unique to this population as sampled (Fig. 2).i> and in Figure 3 where the gAHB individuals form a very distinct cluster (green dots).

Another alternative hypothesis would be that resource constraints lead to a gentle phenotyp and if those gAHB would be transferred back to the mainland in a more reach environment they would become aggressive again.

2. Figure 2 is complex and hard to understand. For example why are there 3 networks for each genes, do those represent different haplotyp blocks within a gene? But if so why are then different haplotype frequencies shown in part B of the figure and what do the different colors mean, size of haplotype blocks within those genes in differnt populations. Overall, I think this figure could go since it really doesn't convey a convincing picture for the reader about the statistical analysis.

Reviewer #2 (Remarks to the Author):

The conclusions of this paper are similar to those of (Galindo-Cardona et al., 2013), based on microsatellites: the honey bee population of Puerto Rico is the product of hybridization of African Honey Bees (AHB) and European honey bees (EHB) , and admixture rapidly gave way to a homogenous hybrid population. The Puerto Rico population retains more EHB alleles than do AHB populations elsewhere.

Admixture giving way to hybridization has been repeatedly shown in the literature of Africanized bees in the Americas, right back to the early 1990s (e.g. Pinto et al., 2004; Pinto et al., 2005; Rinderer et al., 1991; Sheppard et al., 1991), including more recently with SNP genotyping (Nelson et al., 2017; Wallberg et al., 2014; Zayed and Whitfield, 2008).

The new dimension in this manuscript to identify whole regions that to have been under selection for genes of EHB or AHB origin – though this kind of thing has been done before for the Brazilian population (Nelson et al., 2017/ though this germane paper is not cited). While I appreciate that the criteria for acceptance in Nature Communications is impact on a field rather than more general impact, I doubt that this paper will have substantial impact on the field than (say) Nelson et al. (2017). Maybe the CSS way of looking discovering evidence of selection will get some traction.

The manuscript assumes that the 'original' honey population of Puerto Rico was gentle EHB, that it was then invaded by AHB, which were aggressive. The AHB strain then took over the island, hybridizing as it went, following a similar pattern to central and South America. Then, natural selection changed the PR population back to being gentle, while retaining other characteristics of AHB.

An alternative scenario is that the number of invading AHB was low, and infrequent, and

that aggression was lost by dilution and genetic drift, rather than by natural selection. Its important that your paper establishes complete hybridization with limited admixture across the island.

There is a disconnect between the genomics and the assumption that the PR population has been made more gentle by a 'soft' selective sweep. We really don't know if there is any association between gentleness and the genomic regions that have been identified as subject to selection. This is speculation.

Some details

Figure 1. Define CSS in the legend for those that wont read the text carefully.

L422-447 This does not make sense. What you need to do is calculate the probability of homozygotes, which is the sum of the squared frequencies of each *csd* allele. Did you mean to say diploid males rather than diploids? Also, the intention seems to be to calculate the relative frequency of diploid males in sub populations relative to total population. Why is that interesting? I'd rather know the frequency in each population.

Also, the *csd* analysis described here does not seem to appear in the Results.

L 37. This is not accurate. The introduction was not accidental, it was deliberate (e.g. Spivak et al., 1991).

L 42. Health crisis is over stated. While not to minimize that there were deaths, the main impacts were changes to beekeeping practice.

Galindo-Cardona A, Acevedo-Gonzalez JP, Rivera-Marchand B, Giray T, 2013. Genetic structure of the gentle Africanized honey bee population (gAHB) in Puerto Rico. *Bmc Genetics* 14. doi: 10.1186/1471-2156-14-65.

Nelson RM, Wallberg A, Simões ZLP, Lawson DJ, Webster MT, 2017. Genomewide analysis of admixture and adaptation in the Africanized honeybee. *Mol Ecol* 26:3603-3617. doi: 10.1111/mec.14122.

Pinto MA, Rubink WL, Coulson RN, Patton JC, Johnston JS, 2004. Temporal pattern of Africanization in a feral honeybee population from Texas inferred from mitochondrial DNA. *Evolution* 58:1047-1055.

Pinto MA, Rubink WL, Patton JC, Coulson RN, Johnston JS, 2005. Africanization in the United States: Replacement of feral European honeybees (*Apis mellifera* L.) by an African hybrid swarm. *Genetics* 170:1653-1665.

Rinderer TE, Stelzer JA, Oldroyd BP, Buco SM, Rubink WL, 1991. Hybridization between European and Africanized honey bees in the neotropical Yucatan peninsula. *Science* 253:309-311.

Sheppard WS, Rinderer TE, Mazzoli JA, Stelzer JA, Shimanuki H, 1991. Gene flow between African- and European-derived honey bee populations in Argentina. *Nature* 349:782-784.

Spivak M, Fletcher DJC, Breed MD, 1991. Introduction. In: Spivak M, Fletcher DJC, Breed MD, editors. *The "African" honey bee* Boulder: Westview Press. p. 1-9.

Wallberg A, Han F, Wellhagen G, Dahle B, Kawata M, Haddad N, Simoes ZLP, Allsopp MH, Kandemir I, De la Rua P, Pirk CW, Webster MT, 2014. A worldwide survey of genome

sequence variation provides insight into the evolutionary history of the honeybee *Apis mellifera*. Nature Genetics advance online publication. doi: 10.1038/ng.3077
<http://www.nature.com/ng/journal/vaop/ncurrent/abs/ng.3077.html> - supplementary-information.

Zayed A, Whitfield CW, 2008. A genome-wide signature of positive selection in ancient and recent invasive expansions of the honey bee *Apis mellifera*. Proc Nat Acad Sci USA 105:3421-3426.

Reviewer #3 (Remarks to the Author):

This is a very interesting, clever, and informative study. The authors take advantage of three natural populations of honey bees that differ in aggression - the European (EHB; domesticated and docile), the African (AHB; more "wild" and more aggressive) and an island population of the African form (gAHB) that is now less aggressive, resembling EHB. This "triangulation" as the authors call it allows them to make specific a priori predictions regarding genetic change as a result of selection - drift might explain AHB to gAHB changes if they only had the pairwise comparison. This is well set up.

Frankly, I have little to complain about in the paper. I might have come up with alternative explanations - I don't find the human/honey bee story on PR to be compelling, but frankly who am I to say? It is an idea. It is presented as an idea. That is the point of writing scientific papers - to present new ideas as well as to test hypotheses. This is what makes it worthy of Nature Communication. Its not a report. Its a paper.

I might quibble with a few details, but I don't see the point. This paper tells a complete story, is well supported by data and analysis. The question is compelling. I fully support publication as I think this is a topic of tremendous interest and a cool system. I think the work presented here may stimulate others to consider how isolated populations change in behavior and how that might reflect underlying genetic changes.

We would like to thank the reviewers for their thoughtful comments. Responses to each comment are given below, along with references to the revised manuscript:

Reviewers' comments:

Reviewer #1 (Remarks to the Author):

General Comments:

This is an excellent study that uses a extremely large data set (90 independent genomes from three different geographic locations). The combination of F_{ST} and haplotype homozygosity (as a measure for selection) helped identify many candidat regions and genes. I recommend the publication of this manuscript but would ask the authors to include/discuss alternative explanations (see below).

The presentation and documentation of their methods is very good.

Specific comments:

1. I am not sure that the author could rule out an alternative scenario to their proposed hard/soft sweep, namely, colonization by a founding population that was exclusively gAHB or had a high frequency of gAHB genotypes already.

I assume the author mean selection (see below) by humans in an urban environment.

Selection likely involved the elimination of the most aggressive of the invasive AHB colonies.

147 With reduced competition for floral resources, gentler colonies would have increased reproductive 148 success.

This didn't happen in any other population where a lot of africanized feral bees are still out-competing the highly managed EHB populations. Is there anything known about the composition of the feral africanized bees. If those are also gentle an alternative explanation would be that the initial colonizers were already gentle or gentle AHB were colonizing the island at a high frequency and this trait was fixed by genetic drift.

In particular, looking back at the published records on the rapid decrease in aggressiveness seems not that conclusive, i.e. the evidence is based on public media reports rather than hard scientific evidence.

Marchand et al. 2012:

In preliminary observations, the Africanized bees of Puerto Rico seemed to be less defensive than expected while retaining their typical size and mite resistance. Moreover, a dramatic decrease in reported attacks since the arrival of Africanized bees in Puerto Rico (Rivera-Marchand 2006) supports the hypothesis of decreased defensiveness. For instance, there had been four deaths in the first 4 years of the arrival of Africanized bees and none in the last 10 years. This pattern is significant in a simple runs test. The number of incidents where bees attacked people, as investigated in newspaper archives, shows a 100-fold decrease in numbers in the same time period. These data may indicate reduced defensiveness in Africanized bees from Puerto Rico, although public awareness could account for part of the decrease in incidents (Rivera-Marchand 2006).

Finally, additional evidence for a unique composition of the island population is given by the authors <In addition, some

102 haplotypes present at high frequencies in gAHB are unique to this population as sampled (Fig. 2).i> and in Figure 3 where the gAHB individuals form a very distinc cluster (green dots).

Another alternative hypothesis would be that resource constraints lead to a gentle phenotyp and if those gAHB would be transferred back to the mainland in a more reach environment they would become aggressive again.

Response:

We thank the reviewer for their supportive and insightful observations. We agree that discussions of alternative hypothesis were lacking, and have incorporated this into the revised manuscript (see lines 198-212).

We appreciate the reviewer's observation on the uniqueness of gentleness within gAHB. The honey bees (feral and managed) currently found across PR represent one contiguous population from an AHB background (see Results section "Relationships between gAHB and other honey bee populations", lines 135-146 of the revised manuscript). Morphometry-based typing of the population was presented by Rivera-Marchand et al., 2006, and a marker-based profiling of the population is provided in the work by Galindo-Cardona et al., 2013. We also show that the Puerto Rico population is genetically AHB (Figure 3) and consistent with North American origin.

The invasive population was described as aggressive initially (even if there were no careful behavioral studies at the time, the public alarm, hospital records and deaths from stings, as seen in other AHB invasions, provide strong circumstantial evidence that this was the case). As no gentle AHB populations are known on any of the potential mainland sources of the population, it is hard to conclude anything other than that the invasive population was aggressive just like the source populations. Thus, we conclude that the gentle phenotype must have evolved on the island, and have modified the manuscript to reflect this line of reasoning (lines 198-212).

We hypothesize that the unique driving factor for the evolution of gentleness in gAHB was unsupervised selection by humans during the initial founding period aided by unique ecological characteristics of Puerto Rico as an oceanic island. In the revised manuscript, we have expanded our reasoning for this hypothesis to incorporate greater detail (see lines 168-192).

We also thank the reviewer for their very interesting observation on the effect of resource constraints. This is a point that we had not previously thought of, and is consistent with a soft sweep which retains aggressive alleles in the population. We thank the reviewer for bringing this up, and include a discussion of this hypothesis in our revised manuscript (lines 213-219).

2. Figure 2 is complex and hard to understand. For example why are there 3 networks for each genes, do those represent different haplotyp blocks within a gene? But if so why are then different haplotype frequencies shown in part B of the figure and what do the different colors mean, size of haplotype blocks within those genes in differnt populations. Overall, I think this figure could go since it really doesn't convey a convincing picture for the reader about the statistical analysis.

Response:

We agree that the original figure was confusing. We have modified Figure 2 to make it more convincing and clear according to the reviewer's suggestions. The revised figure shows clearly the different haplotype composition of the three populations.

Reviewer #2 (Remarks to the Author):

The conclusions of this paper are similar to those of (Galindo-Cardona et al., 2013), based on microsatellites: the honey bee population of Puerto Rico is the product of hybridization of African Honey Bees (AHB) and European honey bees (EHB), and admixture rapidly gave way to a homogenous hybrid population. The Puerto Rico population retains more EHB alleles than do AHB populations elsewhere.

Response:

We agree that there are certainly some AHB populations with a lower EHB allele frequency than gAHB, and discuss this in the manuscript. Our population is more like the EHB from the M group than other AHB sampled. However, our Figure 3 clearly shows that there are some AHB colonies in the Mexican spectrum that are more EHB-like than gAHB. We have clarified this interpretation in the revised manuscript (see lines 198-212).

Admixture giving way to hybridization has been repeatedly shown in the literature of Africanized bees in the Americas, right back to the early 1990s (e.g. Pinto et al., 2004; Pinto et al., 2005; Rinderer et al., 1991; Sheppard et al., 1991), including more recently with SNP genotyping (Nelson et al., 2017; Wallberg et al., 2014; Zayed and Whitfield, 2008).

Response:

We thank the reviewer for their comment and agree that the process of admixture leading to hybridization in AHB has been extensively studied. However, none of these studies showed selection on a hybridized population leading to a gentle phenotype. Our intent was to frame how known processes of admixture in AHB likely contributed to the genetic diversity of the founding gAHB colony, evident in the relationship between haplotypes (Fig. 2b), and allowed this population to evolve a gentle phenotype. We have tried to make this distinction clearer in the revised manuscript, including full reference to the prior AHB admixture studies (lines 143-146, 198-212).

The new dimension in this manuscript to identify whole regions that to have been under selection for genes of EHB or AHB origin – though this kind of thing has been done before for the Brazilian population (Nelson et al., 2017/ though this germane paper is not cited). While I appreciate that the criteria for acceptance in Nature Communications is impact on a field rather than more general impact, I doubt that this paper will have substantial impact on the field than (say) Nelson et al. (2017). Maybe the CSS way of looking discovering evidence of selection will get some traction.

Response:

We now cite the Nelson et al., 2017 paper, which came out shortly before our manuscript was submitted (see line 143). A key difference from Nelson et al. 2017 is that we identify regions under selection during the evolution of a unique gentle population. We hope that this is clearer in our revision.

The manuscript assumes that the 'original' honey population of Puerto Rico was gentle EHB, that it was then invaded by AHB, which were aggressive. The AHB strain then took over the island, hybridizing as it

went, following a similar pattern to central and South America. Then, natural selection changed the PR population back to being gentle, while retaining other characteristics of AHB.

An alternative scenario is that the number of invading AHB was low, and infrequent, and that aggression was lost by dilution and genetic drift, rather than by natural selection. Its important that your paper establishes complete hybridization with limited admixture across the island.

Response:

As noted in the response to Reviewer #1, this is a plausible scenario, but unlikely given that the gAHB population remains genetically AHB and is distinct from known EHB populations. We have incorporated discussion of this alternative scenario in the revised manuscript (see lines 198-212).

There is a disconnect between the genomics and the assumption that the PR population has been made more gentle by a 'soft' selective sweep. We really don't know if there is any association between gentleness and the genomic regions that have been identified as subject to selection. This is speculation.

Response:

The reviewer is completely correct. We no longer assert any functional connection between specific genotypes and the gentle phenotype, only that the genotypes were under selection during the evolution of the gentle AHB. We have removed the offending text in the revised manuscript (lines 67-69).

Some details

Figure 1. Define CSS in the legend for those that wont read the text carefully.

Response:

We now define the abbreviation in the legend.

L422-447 This does not make sense. What you need to do is calculate the probability of homozygotes, which is the sum of the squared frequencies of each *csd* allele. Did you mean to say diploid males rather than diploids? Also, the intention seems to be to calculate the relative frequency of diploid males in sub populations relative to total population. Why is that interesting? I'd rather know the frequency in each population.

Also, the *csd* analysis described here does not seem to appear in the Results.

Response:

We are calculating a likelihood metric for the diploid males in the next generation, not the frequency metric on the current population that the reviewer suggests. Our metric is a direct test whose only assumption is that the male haplotypes sampled are representative of the distribution of mating haplotypes in the population. The result using either method is ultimately the comparison of diploid male frequencies between the populations. Our conclusion is that diversity in gAHB at the *csd* locus is consistent with a bottleneck, and therefore balancing selection at the *csd* locus is an unlikely driver for the evolution of gAHB or the retention of genetic diversity. The

finding was originally provided in the Supplemental Information not Results, as it confirms what our other cluster analysis had already determined. We have moved the results and discussion of this analysis to the main text of the manuscript since the reviewer considers it to be sufficiently important, and have elaborated on the implications of the method used (see Results section “Analysis of population diversity and selection using the sex determination locus”, lines 113-133 in the revised manuscript). In addition, in the original manuscript we had used “diploids” rather than repeating “diploid males”; we now use “diploid males” exclusively for clarity.

L 37. This is not accurate. The introduction was not accidental, it was deliberate (e.g. Spivak et al., 1991).

Response:

We thank the reviewer for pointing out that we were not precise enough in our language. The reviewer is indeed correct that *A. m. scutellata* was intentionally brought to Brazil to produce a more tropical-adapted managed honeybee. We meant in our original manuscript to reflect the unintentional introduction into the greater Brazilian countryside when swarms from aggressive hives escaped confinement. We have modified our language to make the history clearer (Spivak et al., 1991; lines 36-38).

L 42. Health crisis is over stated. While not to minimize that there were deaths, the main impacts were changes to beekeeping practice.

Response:

We have modified the text accordingly, simply noting that there have been “several deaths and widespread public concern” (lines 41-42).

Galindo-Cardona A, Acevedo-Gonzalez JP, Rivera-Marchand B, Giray T, 2013. Genetic structure of the gentle Africanized honey bee population (gAHB) in Puerto Rico. *Bmc Genetics* 14. doi: 10.1186/1471-2156-14-65.

Nelson RM, Wallberg A, Simões ZLP, Lawson DJ, Webster MT, 2017. Genomewide analysis of admixture and adaptation in the Africanized honeybee. *Mol Ecol* 26:3603-3617. doi: 10.1111/mec.14122.

Pinto MA, Rubink WL, Coulson RN, Patton JC, Johnston JS, 2004. Temporal pattern of Africanization in a feral honeybee population from Texas inferred from mitochondrial DNA. *Evolution* 58:1047-1055.

Pinto MA, Rubink WL, Patton JC, Coulson RN, Johnston JS, 2005. Africanization in the United States: Replacement of feral European honeybees (*Apis mellifera* L.) by an African hybrid swarm. *Genetics* 170:1653-1665.

Rinderer TE, Stelzer JA, Oldroyd BP, Bucu SM, Rubink WL, 1991. Hybridization between European and Africanized honey bees in the neotropical Yucatan peninsula. *Science* 253:309-311.

Sheppard WS, Rinderer TE, Mazzoli JA, Stelzer JA, Shimanuki H, 1991. Gene flow between African- and European-derived honey bee populations in Argentina. *Nature* 349:782-784.

Spivak M, Fletcher DJC, Breed MD, 1991. Introduction. In: Spivak M, Fletcher DJC, Breed MD, editors. *The "African" honey bee* Boulder: Westview Press. p. 1-9.

Wallberg A, Han F, Wellhagen G, Dahle B, Kawata M, Haddad N, Simoes ZLP, Allsopp MH, Kandemir I, De la Rua P, Pirk CW, Webster MT, 2014. A worldwide survey of genome sequence variation provides insight into the evolutionary history of the honeybee *Apis mellifera*. *Nature Genetics* advance online publication. doi: 10.1038/ng.3077

<http://www.nature.com/ng/journal/vaop/ncurrent/abs/ng.3077.html> - supplementary-information.

Zayed A, Whitfield CW, 2008. A genome-wide signature of positive selection in ancient and recent invasive expansions of the honey bee *Apis mellifera*. Proc Nat Acad Sci USA 105:3421-3426.

Response:

We thank the reviewer for providing these references, and with the above revisions have added key references in this list to the manuscript.

Reviewer #3 (Remarks to the Author):

This is a very interesting, clever, and informative study. The authors take advantage of three natural populations of honey bees that differ in aggression - the European (EHB; domesticated and docile), the African (AHB; more "wild" and more aggressive) and an island population of the African form (gAHB) that is now less aggressive, resembling EHB. This "triangulation" as the authors call it allows them to make specific a priori predictions regarding genetic change as a result of selection - drift might explain AHB to gAHB changes if they only had the pairwise comparison. This is well set up.

Frankly, I have little to complain about in the paper. I might have come up with alternative explanations - I don't find the human/honey bee story on PR to be compelling, but frankly who am I to say? It is an idea. It is presented as an idea. That is the point of writing scientific papers - to present new ideas as well as to test hypotheses. This is what makes it worthy of Nature Communication. It's not a report. It's a paper.

I might quibble with a few details, but I don't see the point. This paper tells a complete story, is well supported by data and analysis. The question is compelling. I fully support publication as I think this is a topic of tremendous interest and a cool system. I think the work presented here may stimulate others to consider how isolated populations change in behavior and how that might reflect underlying genetic changes.

Response:

We thank the reviewer for the interest in our study and the strongly supportive comments.

REVIEWERS' COMMENTS:

Reviewer #1 (Remarks to the Author):

General Comments:

Overall, I think this is an impressive study and I think it is worthwhile to be published in Nature Communication. The authors have responded to all of my criticism sufficiently. However, after re-reading the manuscript and the comments of the other reviewers I have still a couple of questions and comments. If addressed appropriately

The claim in Line 87-89 contrasts with the stronger claim in the title of the paper (see below). I can fully support the more tentative claim but the claim in the title is in my opinion not supported by data simply because the genetic architecture of gentle behavior is not clear, i.e. I have seen not proof that any of the genes under positive selection are actually causing a more gentle behavior! Additionally, a soft selective sweep is most broadly defined as a selective sweep at multiple loci. I think the authors should that multiple unlinked loci have swept to fixation it is again not clear whether and how many of these are linked to gentle behavior. Their data could still be consistent with one locus of major effect on aggression! Hence I would suggest to modify the title to reflect what is written in the rest of the paper!

L87-89: ...that may be associated with the evolution of gentle behavior...

Rapid evolution of behaviour in a gentle Africanized honeybee by soft selective sweep

The strong bottleneck effect, demonstrated by the *csd* analysis also casts some doubt on using *F_{ST}* values to determine selection in a population. Genetic drift alone can generate extreme *F_{ST}* values in recent invasion events with strong bottleneck effects. Maybe the authors could discuss this alternative explanation as well.

L 148-151: At some point there had to be some admixture in particular at the loci that show strong EHB haplotype signatures (L72). So, even if there was no general admixture signature over the whole genome there was strong admixture or rather introgression from the local EHB population into the gAHB population. That means the gentle behavior has evolved on the island but the gentleness gene/s have been introgressed from the resident EHB population and the new gAHB has then either outperformed both EHB and AHB population on the island. It is not clear why gAHB colonies have a higher fitness than gAHB.

Reviewer #2 (Remarks to the Author):

I am happy with the revisions and recommend publication.

REVIEWERS' COMMENTS:

Reviewer #1 (Remarks to the Author):

General Comments:

Overall, I think this is an impressive study and I think it is worthwhile to be published in Nature Communication. The authors have responded to all of my criticism sufficiently. However, after re-reading the manuscript and the comments of the other reviewers I have still a couple of questions and comments. If addressed appropriately

The claim in Line 87-89 contrasts with the stronger claim in the title of the paper (see below). I can fully support the more tentative claim but the claim in the title is in my opinion not supported by data simply because the genetic architecture of gentle behavior is not clear, i.e. I have seen not proof that any of the genes under positive selection are actually causing a more gentle behavior! Additionally, a soft selective sweep is most broadly defined as a selective sweep at multiple loci. I think the authors should that multiple unlinked loci have swept to fixation it is again not clear whether and how many of these are linked to gentle behavior. Their data could still be consistent with one locus of major effect on aggression! Hence I would suggest to modify the title to reflect what is written in the rest of the paper!

L87-89: ...that may be associated with the evolution of gentle behavior...

Rapid evolution of behaviour in a gentle Africanized honeybee by soft selective sweep

We thank the reviewer for their interest in our study, and agree that the claim in our title may be overstated. In the revised draft of the manuscript we have altered the title (and a section title) to one we feel more accurately describes the co-occurrence of the observed soft selective sweep and change in behavior.

The strong bottleneck effect, demonstrated by the *csd* analysis also casts some doubt on using F_{ST} values to determine selection in a population. Genetic drift alone can generate extreme F_{ST} values in recent invasion events with strong bottleneck effects. Maybe the authors could discuss this alternative explanation as well.

We agree that F_{ST} analysis would be insufficient, so we also used an *Rsb* signal to conclude that we are observing selection. We apologize if this was not clear and have expanded our report of both the F_{ST} and *Rsb* results which we hope provides a better explanation. (see *Signatures of selections* subheading in Results).

L 148-151: At some point there had to be some admixture in particular at the loci that show strong EHB haplotype signatures (L72). So, even if there was no general admixture signature over the whole genome there was strong admixture or rather introgression from the local EHB population into the gAHB population. That means the gentle behavior has evolved on the island but the gentleness gene/s have been introgressed from the resident EHB population and the new gAHB has then either outperformed both EHB and AHB population on the island. It is not clear why gAHB colonies have a higher fitness than gAHB.

We think that the reviewer is asking for a better explanation of how the EHB alleles could have become introgressed into gAHB (they can originate from AHB as well as EHB since the AHB founder population was already admixed). We have modified the text in an attempt to make this clearer.

Reviewer #2 (Remarks to the Author):

I am happy with the revisions and recommend publication.

We thank the reviewer and are glad we could address the prior concerns.